# Adsorption and Gas-Sensing Properties of Ag_n_ (n = 1–4) Cluster Doped GeSe for CH_4_ and CO Gases in Oil-Immersed Transformer

**DOI:** 10.3390/nano12234203

**Published:** 2022-11-26

**Authors:** Aijuan Dong, Meiling Sun, Yingang Gui

**Affiliations:** 1Qinhuangdao Vocational and Technical College, Qinhuangdao 066100, China; 2College of Engineering and Technology, Southwest University, Chongqing 400715, China

**Keywords:** Ag_n_-GeSe, adsorption, DFT, CH_4_, CO

## Abstract

The adsorption mechanism of CO and CH_4_ on GeSe, modified with the most stable 1–4 Ag-atom clusters, is studied with the help of density functional theory. Adsorption distance, adsorption energy, total density of states (TDOS), projected density of states (PDOS), and molecular orbital theory were all used to analyze the results. CO was found to chemisorb exothermically on GeSe, independent of Ag cluster size, with Ag_4_-GeSe representing the optimum choice for CO gas sensors. CH_4_, in contrast, was found to chemisorb on Ag-GeSe and Ag_2_-GeSe and to physisorb on Ag_3_-GeSe and Ag_4_-GeSe. Here, Ag GeSe was found to be the optimum choice for CH_4_ gas sensors. Overall, our calculations suggest that GeSe modified by Ag clusters of different sizes could be used to advantage to detect CO and CH_4_ gas in ambient air.

## 1. Introduction

The oil-immersed transformer has been widely used in the modern power system due to its low cost and high power conversion efficiency [1]. Transformer insulation oil is mainly alkanes, cycloalkanes, saturated hydrocarbons, aromatic unsaturated hydrocarbons, and their compounds [2,3,4]. However, during a long service period, a transformer may inevitably suffer from local overheating and partial discharge faults [5,6]. These faults threaten the safety of the entire electrical system because the huge heat and strong distorted electrical field released by the faults may lead to the rupture of the C-C bond and C-H bond of the insulating oil medium, resulting in the generation of activated hydrogen and unstable hydrocarbon free radicals [7,8,9]. With the existence of impurities (H_2_O and O_2_) in the transformer, various decomposition products are dissolved in oil products, such as H_2_, CH_4_, CO, CO_2_, C_2_H_2_, and C_2_H_6_ [10,11,12]. Decomposition product detection is an effective method for online monitoring of transformer faults [13,14]. Since CO and CH_4_ are two typical gases in transformer faults, the condition of a transformer can be predicted by analyzing the concentrations of these two gases [15,16,17]. Due to the low cost and portability of gas sensors, it has been widely used in various fields, including electric power online monitoring [18,19]. Therefore, the gas sensor-based detection of CO and CH_4_ could be a potentially effective means to realize fault detection in transformers [20].

In recent years, GeSe has been widely used in gas-sensing materials because it has a large specific surface area and abundant hole structure [21,22]. More resistant to oxidation and more stable at high temperatures than carbon nanotubes, GeSe is therefore more suitable for gas detection than carbon nanotubes [23,24]. As a result, it is one of the most widely used materials in high temperature and high pressure environments [25]. Gui et al. studied the adsorption behavior of CO, CH_4_, C_2_H_2_, C_2_H_4_ on metal oxide (CuO, NiO, Ag_2_O)-doped GeSe surfaces; Guo et al. investigated the adsorption characteristics of C_2_H_2_, CH4_4_, H_2_ on SnO_2_-GeSe (SnO_2_ doped onto GeSe surfaces) [26,27]. However, the ability of intrinsic GeSe to adsorb gases is limited, such as CH_4_, C_2_H_2_, H_2_, etc., [12,28]. Doping of metal clusters is a common method of material surface modification, which improves gas detection accuracy and adsorption capacity by changing the energy gap of gas sensors. [29,30]. Among the most used metal clusters (Pd, Pt, Au, Ag, Ni) and other metal elements, doping brings good adsorption characteristics and adsorption capacity [31,32,33,34,35,36]. Silver metal has been widely studied in doping modification because of its good physical properties. W.A et al.’s application of Ag doping shows that Ag has a good modified adsorption function and can bring better adsorption capacity to the substrate [37]. In this paper, GeSe was doped with Ag clusters to enhance the gas-sensitive response to CH_4_ and CO.

In this paper, the Ag cluster-modified GeSe is proposed as a promising sensing material for monitoring transformer faults. First, the most stable doping structures of 1–4 Ag atoms on GeSe (Ag_n_-GeSe) were optimized. Then, the most stable structures were chosen to adsorb the gas molecules. The Ag clusters’ doping and gas adsorption mechanisms were analyzed. By analyzing the structure optimization, adsorption energy, band gap, the density of states (DOS), and charge transfer, it was found that the Ag cluster-modified GeSe sensor with high detection response and detection speed has great potential to become a new type of resistive gas sensor. This study provides a new monitoring method and way for the gas insulation monitoring of decomposing components in oil.

## 2. Computational Details and Methods

All calculations were performed based on the density functional theory (DFT) [33,38]. A generalized gradient approximation (GGA) was used to calculate the electron exchange and correlation energy [39,40]. The Perdew–Burke–Ernzerhof (PBE) function was used to calculate the interaction effect between electrons [41]. The DFT-based semi-core pseudopotential (DSPP) and double numerical plus polarization (DNP) were selected [42]. The self-consistent field convergence precision was set to 1 × 10^−6^ Ha. The energy convergence accuracy, maximum stress, and max displacement were set as 1 × 10^−5^ Ha, 2 × 10^−3^ Ha/Å, and 5 × 10^−3^ Ha, respectively [43,44]. Since Ag_n_-GeSe is not magnetic, the spin polarization is limited during structural optimization [45]. A k-point grid of 5 × 5 × 1 was selected for the Brillouin zone integration to obtain accurate energies and structures [46]. In the solvent model, the dielectric constant was set to 2.2 to simulate the insulating oil model. To avoid layer-to-layer interactions, we constructed a 4 × 4 × 1 single nanotube supercell containing 32 Se atoms and 32 Ge atoms for DFT calculations. The distance between layers was greater than 25 Å.

As defined in Equation (1), the adsorption energy represents the energy change in the adsorption process and can be analyzed to find the most stable structure of gas adsorption. If the adsorption energy is negative, it means that the reaction is exothermic and occurs spontaneously. Charge transfer (*Q_T_*) was obtained by Mulliken population analysis. As defined in Equation (2), *Q_ads_* and *Q_iso_*, respectively, represent the net carried charges of the gas molecule after and before adsorption. A positive *Q_T_* shows that electrons transfer from the gas molecule to the Ag_n_-GeSe monolayer. The energy gap between the highest occupied orbit (HOMO) and the lowest unoccupied orbit (LUMO) is defined in Equation (3). The total density of states (TDOS) and projected density of states (PDOS) were analyzed in detail to analyze the mechanism of the adsorption process.
*E_ads_ = E_Agn-GeSe/gas_ − E_Agn-GeSe_ − E_gas_*(1)
*Q_T_ = Q_ads_ − Q_iso_*(2)
*E_g_ = |E_HOMO_ − E_LUMO_*|(3)

## 3. Results and Discussion

### 3.1. Geometry Optimization

In order to study the gas adsorption characteristics of GeSe, the top view and side view of GeSe were obtained through modeling, as shown in Figure 1. The structures of CO and CH_4_ molecules were obtained as shown in Figure 1c,d. The bond lengths between Ge and Se were 2.543 Å (longitudinal) and 2.612 Å (transverse). The reason why the transverse distance is larger than the longitudinal distance is that Ge bonds with two transverse Se atoms, while the longitudinal one bonds with only one Se atom, making the longitudinal Se atom more stable and the longitudinal bond length shorter. The CO molecule is a linear structure with a bond length of only 1.142 Å. The CH_4_ molecule is a regular tetrahedral structure, and it is a stable gas molecule in air. The bond length of each C-H bond is 1.096 Å, and the bond angle is 109.480°.

Figure 2 shows the most stable structure of Ag_n_-GeSe obtained by doping one to four Ag atoms. The doping distance is 2.499 Å, 2.539 Å, 2.680 Å, and 2.702 Å for 1–4 Ag atoms modified GeSe. Based on the Mulliken population, the four types of Ag cluster act as electron acceptors obtaining 0.048 *e*, 0.184 *e*, 0.206 *e*, and 0.288 *e* electron from GeSe, respectively. The redistribution of electric charge leads to the change of conductivity of the system.

As shown in Figure 3, TDOS and PDOS were analyzed to further analyze the doping mechanism of Ag atom doping on GeSe. The peak values of the TDOS of the four Ag cluster-doped GeSe bases shift to the left obviously, which makes the Fermi level continuous. Figure 3(a2–d2) shows the PDOS of GeSe doped with four types Ag_n_-GeSe. The analysis of PDOS showed that the peak value above the Fermi level shifted to the left due to the hybridization of Ag-4*d*, Se-4*p*, and Ge-4*p* orbits, thus improving the conductivity of the system. It can be seen from Figure 3(a1,a2) that the hybridization of Ag-4*d* and Se-4*p* orbits in one Ag atom doping system from −4.0 eV to −6.0 eV resulted in a significant increase in TDOS at −5.0 eV. It can be seen from Figure 3(b1,b2) that the Ag-4*d* and Se-4*p* orbits of double Ag atoms doping system hybridized at −1.0 eV~−2.0 eV, resulting in a significant increase in TDOS at −1.5 eV. Figure 3(c1,c2) shows that the Ag-4*d* and Se-4*p* orbits hybridized at −3.0 eV~−4.0 eV in the triple Ag atoms doping system, resulting in a significant increase in TDOS at −4.0 eV. It can be seen from Figure 3(d1,d2) that the hybridization of Ag-4*d* and Ge-4*p* orbits of the quadruple Ag atoms doping system at −4.0 eV~−5.0 eV resulted in a significant increase in TDOS at −4.5 eV. In general, a strong orbital hybridization results in a stable Ag_n_-GeSe structure, indicating that Ag cluster-doping on the GeSe surface is stable enough for further gas adsorption.

### 3.2. Analysis of CO Gas Adsorption on Ag_n_-GeSe Surface

To study the adsorption behavior of gas molecules on Ag_n_-GeSe, gas molecules were made to approach Ag atoms from different directions and angles. The adsorption position with the largest adsorption energy was taken as the most stable adsorption structure, and then the density of states, band structure, and molecular orbit of the adsorption structures was analyzed.

Figure 4 shows the most stable CO adsorption structure. The adsorption distances of CO on the four Ag_n_-GeSe systems were 2.080, 2.157, 2.194, and 2.086 Å respectively, and the C-O bond was not damaged in the adsorption process. It can be seen that the adsorption ability of our Ag_n_-GeSe systems to CO was relatively moderate, which was conducive to the subsequent desorption process, resulting in high sensitivity and reusability of the gas-sensing material. The C atom tends to adsorb on the Ag atom in the CO adsorption process.

Figure 5 shows the DOS analysis of Ag_n_-GeSe before and after CO adsorption. It can be seen from Figure 5(a1–d1) that the peak value of TDOS shifted significantly to the left after gas adsorption, making it continuous at the Fermi level. It can be seen from Figure 5(a1,a2) that Ag-GeSe had a new peak value due to the hybridization of Ag-4*d*, C-2*p*, and O-2*p* from −11.0 to −12.0 eV in the CO adsorption process. In Figure 5(b1,b2), it can be seen that Ag_2_-GeSe had a new peak value due to the hybridization of Ag-4*d*, C-2*p*, and O-2*p* from −10.0 to −11.0 eV during CO adsorption. The peak of Ag_3_-GeSe and Ag_4_-GeSe was roughly the same as that of Ag_2_-GeSe.

The adsorption parameters of CO on the four doping structures are shown in Table 1, including adsorption distance, adsorption energy, and charge transfer. The adsorption energies of the four adsorption structures were −0.177, −0.166, −0.171, −0.193 eV. The charge transfer of the four adsorption structures during the adsorption process was 0.134, 0.105, −0.014, −0.165 *e*. The negative charge transfer indicates that the electron transfers from CO gas to Ag_n_-GeSe, while the positive charge transfer indicates the transfer of electrons from Ag_n_-GeSe to CO gas. From the moderate adsorption distance, large adsorption energy, and charge transfer, Ag_4_-GeSe is more suitable for CO gas adsorption.

### 3.3. Analysis of CH_4_ Gas Adsorption on Ag_n_-GeSe Surface

Figure 6 shows the most stable CH_4_ adsorption structure. The adsorption distances of CH_4_ on 1–4 Ag atom-doped GeSe were 2.778, 2.957, 4.164, and 3.328 Å, respectively. The structure of CH_4_ did not change during the adsorption process. Compared with CO adsorption, the adsorption distance of CH_4_ was much larger. The adsorption distances of Ag_3_-GeSe and Ag_4_-GeSe to CH_4_ reached 4.164 and 3.328 Å, respectively. With such a large adsorption distance, it can be inferred that Ag_3_-GeSe and Ag_4_-GeSe show physical adsorption to CH_4_. Since the C atom is surrounded by four H atoms in the CH_4_ molecular structure, the H atom approaches the substrate in the adsorption process. In the four adsorption processes, C-H bonds elongate due to the effect of H-Ag bonding.

Figure 7 shows the density of states before and after CH_4_ adsorption on Ag_n_-GeSe. After CH_4_ adsorption, the TDOS of the system moved significantly to the left, and the filling of electrons at the Fermi level increased, increasing the probability of electrons crossing the gap from the valence band to the conduction band. Therefore, the conductivity increased after CH_4_ adsorption. The TDOS of the four adsorption structures increased at −7.0, −6.0, −6.5 and −5.0 eV, respectively. This is mainly due to the strong hybridization of Ag-4*d*, H-1*s*, and C-2*p* orbits. It can be seen from Figure 7(a2,b2), that there was a strong chemical bond between CH_4_ and Ag. However, the narrow orbital spike in Figure 7 (c2,d2) indicated that there was no chemical bond between CH_4_ and Ag atoms, but only physical adsorption. The invariance of Ag_3_-GeSe and Ag_4_-GeSe at the Fermi level, and the minimal peak changes at other places also confirmed that the reaction was physical adsorption.

The adsorption parameters of CH_4_ on Ag_n_-GeSe are listed in Table 2, including adsorption distance, adsorption energy, and charge transfer. The adsorption energies of the four structures were −0.158, −0.159, −0.122 and −0.018 eV, respectively. The charge transfers were 0.034, 0.013, −0.068, −0.026 *e*, respectively. The long adsorption distance, small adsorption energy, and small charge transfer confirm that Ag_3_-GeSe and Ag_4_-GeSe are physical adsorptions to CH_4_. Ag-GeSe is more suitable for CH_4_ gas adsorption according to the moderate adsorption distance, large adsorption energy, and moderate charge transfer.

### 3.4. Molecular Orbital Theory Analysis of Gases Adsorption on Ag_n_-GeSe

The behavior of electron distribution in the adsorption process was analyzed by molecular orbital theory. The HOMO and LUMO of the CO and CH_4_ adsorption systems are shown in Figure 8 and Figure 9, respectively. The energy gap between HOMO and LUMO can be a key indicator to evaluate the conductivity of the target structure. Before gas adsorption on Ag_n_-GeSe, HOMO mainly distributed over Ag, indicating that the Ag atom provided electrons to interact with CO and CH_4_ gases as an active site. After CO and CH_4_ adsorption, HOMO changes became more concentrated on Ag, while LUMO became more uniform.

As shown in Table 3, the energy gaps of the four CO adsorption structures were 0.053, 0.037, 0.031 and 0.036 eV, respectively. There was a small HOMO and LUMO distribution of Ag_4_-GeSe on Ag atoms upon CO adsorption, indicating that the electron distribution of the system was uniform, and the moderate band gap indicated that Ag_4_-GeSe was more suitable for CO adsorption. The energy gaps of the four CH_4_ adsorption systems were 0.049, 0.044, 0.046 and 0.025 eV, respectively. After Ag-GeSe adsorbed CH_4_, the band gap increased significantly, which made the conductivity of the system decrease significantly, so the conductivity change of the target system was more obvious. Therefore, Ag-GeSe is more suitable for the gas-sensing of CH_4_.

## 4. Conclusions

In this work, the adsorption behaviors of 1–4 Ag atom-modified GeSe to CO and CH_4_ gases were analyzed based on first principle calculations. The interaction mechanism between Ag_n_-GeSe and the gas molecules was comprehensively investigated by analyzing adsorption structure, the density of states, and molecular orbital theory. All four Ag_n_-GeSe structures chemisorb CO gas, but Ag_4_-GeSe is more suitable for CO gas sensors according to proper adsorption distance, large adsorption energy, and proper charge transfer. Ag-GeSe and Ag_2_-GeSe chemisorb, while Ag_3_-GeSe and Ag_4_-GeSe physisorb CH_4_ gas. Based on the density of states and molecular orbital theory analysis, it can be concluded that Ag-GeSe is more suitable for the detection of CH_4_ gas. Although the adsorption mechanism was slightly different for CO and CH_4_ adsorption on different Ag atom-doping systems, the adsorption capacity was very close. In conclusion, Ag cluster-modified GeSe could be a suitable CO and CH_4_ gas-sensing material for use in the power system.

## Figures and Tables

**Figure 1 nanomaterials-12-04203-f001:**
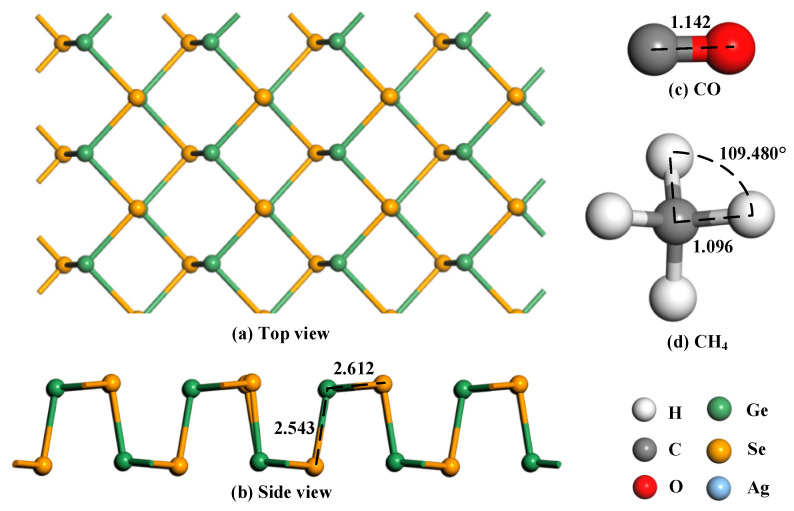
Structures of GeSe and gas molecules (**a**) Top view, (**b**) side view, (**c**) CO molecule, (**d**) CH_4_ molecule. The distance is in Å.

**Figure 2 nanomaterials-12-04203-f002:**
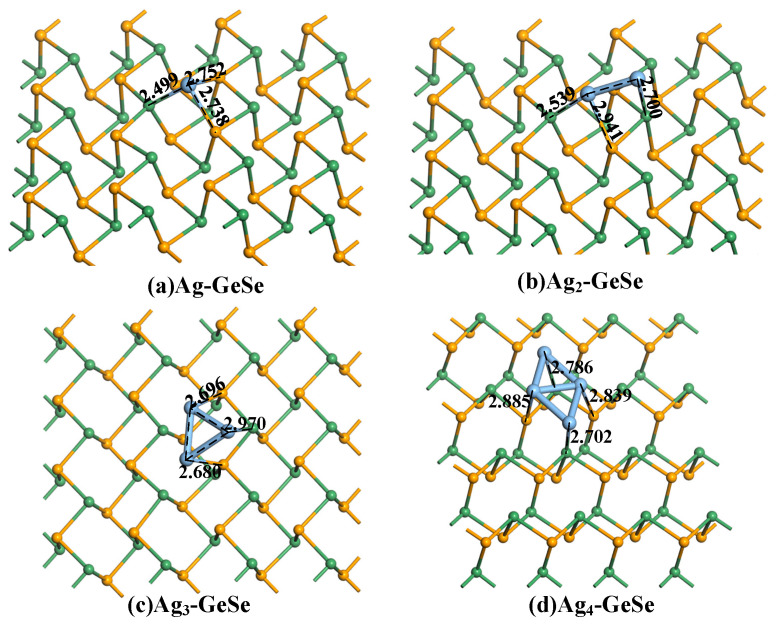
The stable structures of Ag_n_-GeSe. (**a**) Ag-GeSe, (**b**) Ag_2_-GeSe, (**c**) Ag_3_-GeSe, (**d**) Ag_4_-GeSe. The distance is in Å.

**Figure 3 nanomaterials-12-04203-f003:**
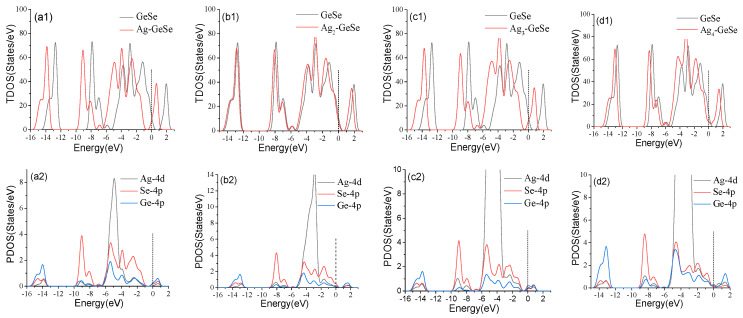
TDOS and PDOS before and after 1~4 Ag atoms doping on GeSe. (**a1**,**a2**) Ag-GeSe, (**b1**,**b2**) Ag_2_-GeSe, (**c1**,**c2**) Ag_3_-GeSe, (**d1**,**d2**) Ag_4_-GeSe.

**Figure 4 nanomaterials-12-04203-f004:**
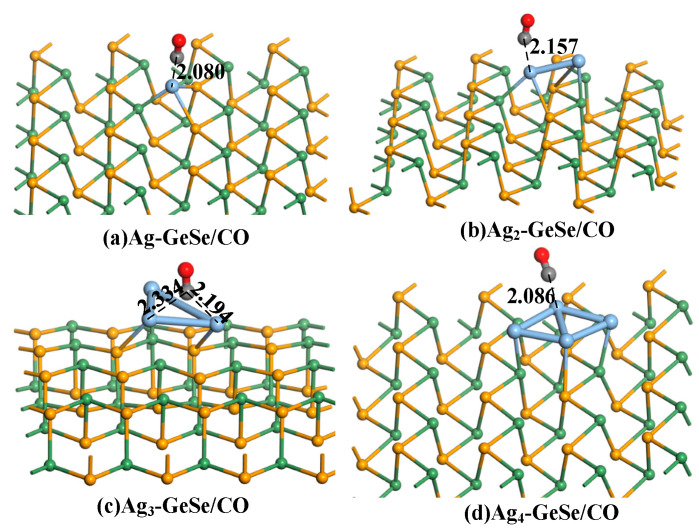
The adsorption structures of gas molecules on Ag_n_-GeSe. (**a**) Ag-GeSe/CO, (**b**) Ag_2_-GeSe/CO, (**c**) Ag_3_-GeSe/CO, (**d**) Ag_4_-GeSe/CO. The distance is in Å.

**Figure 5 nanomaterials-12-04203-f005:**
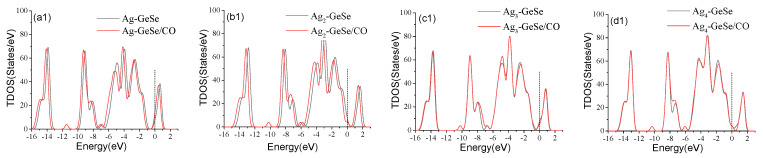
TDOS and PDOS of gas molecules adsorption on Ag_n_-GeSe. (**a1**) TDOS of CO on Ag modified GeSe (**b1**) TDOS of CO on Ag_2_ modified GeSe (**c1**) TDOS of CO on Ag_3_ modified GeSe (**d1**) TDOS of CO on Ag_4_ modified GeSe; (**a2**–**d2**) are PDOS of the corresponding subgraph.

**Figure 6 nanomaterials-12-04203-f006:**
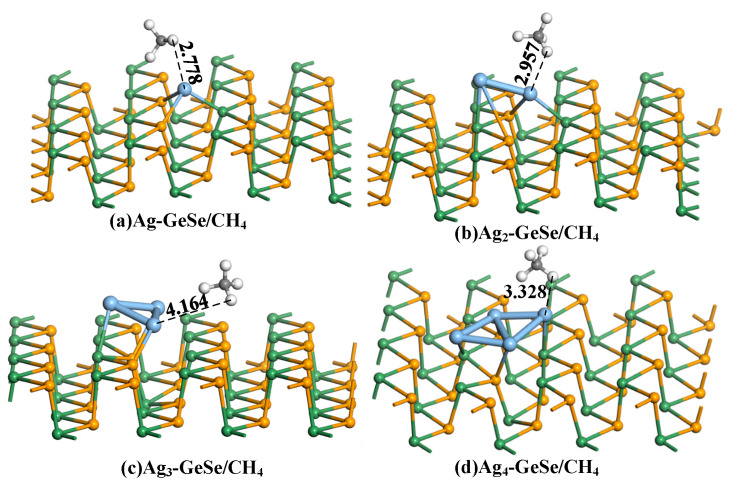
The most stable structures of gas molecules on Ag_n_-GeSe. (**a**) Ag-GeSe/CH_4_ (**b**) Ag_2_-GeSe/CH_4_ (**c**) Ag_3_-GeSe/CH_4_ (**d**) Ag_4_-GeSe/CH_4_.

**Figure 7 nanomaterials-12-04203-f007:**
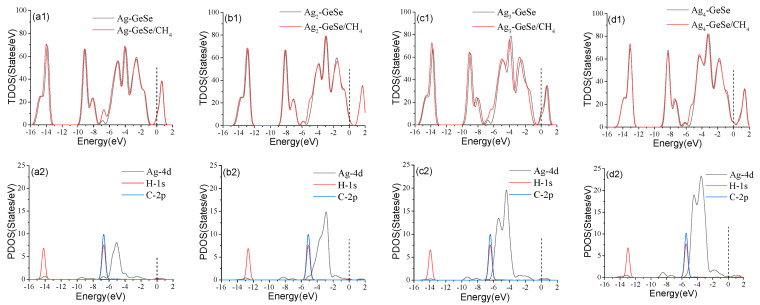
TDOS and PDOS of adsorption of gas molecules on Ag_n_−GeSe. (**a1**) TDOS of CH_4_ on Ag modified GeSe (**b1**) TDOS of CH_4_ on Ag_2_ modified GeSe (**c1**) TDOS of CH_4_ on Ag_3_ modified GeSe (**d1**) TDOS of CH_4_ on Ag_4_ modified GeSe; (**a2**–**d2**) are PDOS of the corresponding subgraph.

**Figure 8 nanomaterials-12-04203-f008:**
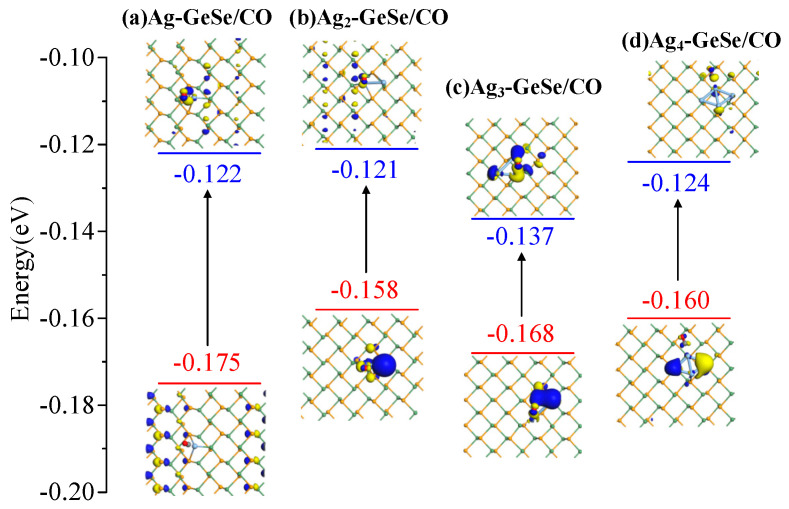
HOMO and LUMO distribution after CO adsorption. (**a**) Ag-GeSe/CO system (**b**) Ag_2_-GeSe/CO system (**c**) Ag_3_-GeSe/CO system (**d**) Ag_4_-GeSe/CO system.

**Figure 9 nanomaterials-12-04203-f009:**
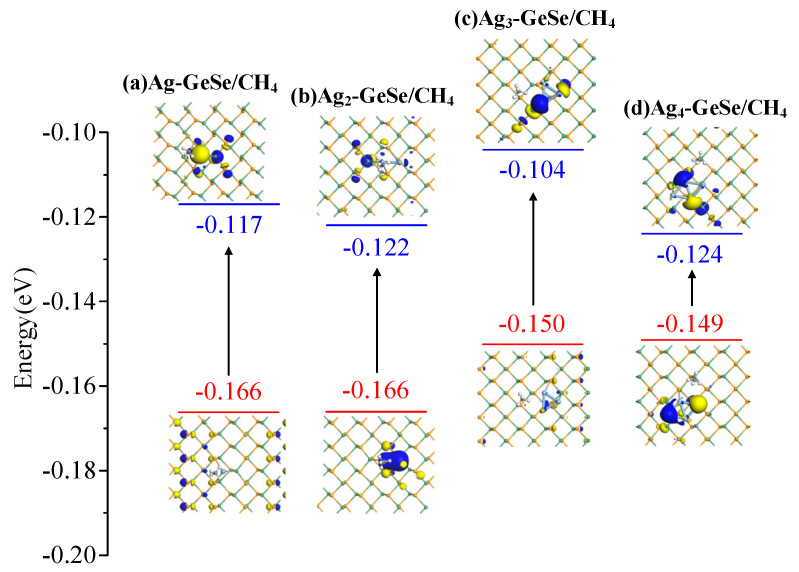
HOMO and LUMO distribution after CH_4_ adsorption. (**a**) Ag-GeSe/CH_4_ system (**b**) Ag_2_-GeSe/CH_4_ system (**c**) Ag_3_-GeSe/CH_4_ system (**d**) Ag_4_-GeSe/CH_4_ system.

**Table 1 nanomaterials-12-04203-t001:** Adsorption parameters of CO gas molecules on Ag_n_-GeSe.

Configuration	Structure	*d* (Å)	*E_ads_* (eV)	*Q_T_* (e)
Ag-GeSe/CO	Figure 4a	2.080	−0.177	0.134
Ag_2_-GeSe/CO	Figure 4b	2.157	−0.166	0.105
Ag_3_-GeSe/CO	Figure 4c	2.194	−0.171	−0.014
Ag_4_-GeSe/CO	Figure 4d	2.086	−0.193	−0.165

**Table 2 nanomaterials-12-04203-t002:** Adsorption parameters of CH_4_ gas molecules on Ag_n_-GeSe.

Configuration	Structure	*d* (Å)	*E_ads_* (eV)	*Q_T_* (e)
Ag-GeSe/CH_4_	Figure 6a	2.778	−0.158	0.034
Ag_2_-GeSe/CH_4_	Figure 6b	2.957	−0.159	0.013
Ag_3_-GeSe/CH_4_	Figure 6c	4.164	−0.122	−0.068
Ag_4_-GeSe/CH_4_	Figure 6d	3.328	−0.018	−0.026

**Table 3 nanomaterials-12-04203-t003:** Energy of HOMO, LUMO, and energy gap of CO and CH_4_ adsorbed Ag_n_-GeSe systems.

Configuration	Structure	*E_HOMO_* (eV)	*E_LUMO_* (eV)	*E_g_* (eV)
Ag-GeSe	\	−4.707	−3.483	1.224
Ag_2_-GeSe	\	−4.555	−3.311	1.244
Ag_3_-GeSe	\	−4.700	−3.648	1.052
Ag_4_-GeSe	\	−4.329	−3.716	0.613
Ag-GeSe/CO Ag_2_-GeSe/CO	Figure 8a Figure 8b	−0.175 −0.158	−0.122 −0.121	0.053 0.037
Ag_3_-GeSe/CO	Figure 8c	−0.168	−0.137	0.031
Ag_4_-GeSe/CO	Figure 8d	−0.160	−0.124	0.036
Ag-GeSe/CH_4_ Ag_2_-GeSe/CH_4_	Figure 9a Figure 9b	−0.166 −0.166	−0.117 −0.122	0.049 0.044
Ag_3_-GeSe/CH_4_	Figure 9c	−0.150	−0.104	0.046
Ag_4_-GeSe/CH_4_	Figure 9d	−0.149	−0.124	0.025

## Data Availability

The data that support the findings of this study are available from the corresponding author upon reasonable request.

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
