# Peer review of "Adsorption and Gas-Sensing Properties of Agn (n = 1–4) Cluster Doped GeSe for CH4 and CO Gases in Oil-Immersed Transformer"

_nanomaterials, 2022, doi:10.3390/nano12234203_

Round 1

Reviewer 1 Report

The manuscript reports a study of Ag atoms on GeSe substrate for CH4 and CO sensing by DFT calculation. The authors build the model of GeSe modified by 1-4 Ag atoms and calculate the CO and CH4 adsorption energy on these structures. The authors conclude that several structures in this study have the potential as CH4 and CO sensor. However, there are some remaining questions before considering publication.

1. For CH4 and CO adsorption, the coverage plays a significant role. What are the coverages used in the study and how is it related to the real sensor application?

2. The authors provide the stabilized Ag-modified GeSe. Will this structure change during the CO/CH4 adsorption?

Reviewer 2 Report

The paper reports on first principles calculations concerning the adsorption of CO and CH4 on GeSe doped with small clusters of Ag. From the application side this study was motivated with the aim of finding Ag(n) GeSe structures which are optimally suited to detect degradation products emerging from degraded transformer oils. 

The paper is well organized, written in decent English and supported by high-quality illustrations. Before publication, however, a number of changes and improvements should be implemented:

Textual changes: please see the attached document.

Other concerns:

For me as a non-expert in DFT calculations - like most other prospective readers - the conclusions concerning th optimum choices of sensor materials for CO (Ag0 GeSe) and CH4 (Ag4 GeSe) are hard to associate with the changes in the TDOS, PDOS and HOMO-LUMO changes. In particular, it is hard to distinguish between physisorption and chemisorption cases in the case of CH4. Likely, these spectra could be explained in amore intuitive way to support non-expert-DFT readers.

Concerning applications:

- Is it technologically feasible to selectively produce GeSe specimens with 1,2, 3 and 4 Ag atoms? I expect that samples will contain mixtures of all kinds of different-size Ag clusters, likely also with many more Ag atoms.

- Based on the present DFT results: can predictions be made concering the CO/CH4 selectivity of the optimum-choice CO and CH4 sensors?

- My guess is that for the degradation monitoring you are proposing two-sensor arrays with one optimum CO and one optimum CH4 sensor. Is this correct?

- I guess the sensors are to be operated in ambient air close to the degraded oil rather than being immersed inside the oil. Is this correct?

- If the sensors are to be operated in ambient air, what about cross senstivties with regard to the largely variable concentrations of water vapour (meteorology), NO2 (automotive exhaust gases) and NH3 (agriculture)?  

-   

Reviewer 3 Report

This paper deals with a theoretical approach to check the viability of using Ag to dope GeSe in order to improve the detection of CH4 and CO. The proposed application is the control quality of oil-immersed transformers.

The first problem arising from this paper is the state of the art revised in the introduction. Although authors cite some references of similar studies published for other materials as MoSe, they do not cite previous works of similar studies performed for GeSe. In fact, there is a similar study for the same application for GeSe doped with CuO, NiO and Ag2O. There is also another publication about the theoretical study for SnO2 doped GeSe also for the same application. Therefore, references should be revised and introduction adapted consequently.

On the other hand, at the end of the introduction, authors claim that the proposed Ag-cluster doped GeSe “can effectively detect faults”, but this fact has not been proved in this paper. Authors have only shown the theoretical possibility of detecting the target gases using the proposed material. Then, some statements must be revised.

Nevertheless, my major concern about this work is about the interest of the study:

Authors propose Ag cluster from 1 atom to 4, analysis for each case which are the effective bonding for the target molecules (CO and CH4), but in a practical situation, how can be controlled the number of atoms for the included cluster? Moreover, since Ag can be easily oxidized, those atoms will be in metallic format or Ag will oxidize in a real situation?

Summarizing, I have my doubts about the interest of this research, but in any case, the introduction must be revised and the possible limitation of this study should be clarified in the conclusions.

Reviewer 4 Report

The article is devoted to quantum-chemical modeling of materials, which are supposed to be used as sensors for methane and carbon monoxide, based on GeSe with 1-4 silver atoms on the surface in the form of a cluster.

Despite the innovation associated with the modification of the GeSe surface with silver atoms, the idea of the work and the approaches used are not original. Similarly, recent works have been carried out and published, such as, for example,

https://doi.org/10.3390/chemosensors10070236

https://doi.org/10.3390/s22103860

However, the key problem of this work is not the lack of originality of the idea or approaches. There are two key problems here, which do not allow me to recommend the work for publication.

1) The first problem is very bad English and a lot of mistakes in terminology. It's hard to list them all because they are ubiquitous. As an example, one can point to the following.

Line 55-58: Authors declare that considered materials demonstrate high detection response and detection speed. However, they did not perform such studies which can show these features. The article is devoted to the electronic structure change of GeSe, modified by Ag atoms, in the process of CO and CH4 molecule adsorption. It is not clear how the statement about high detection response and speed follows from the results.

Lines 58-59: Authors declare that their study provides a new monitoring method and way for gas insulation monitoring of decomposing components in oil, but again it seems strange because no new method is demonstrated in the article. It is about new materials instead, but not the method.

Lines 62-64: It seems following from the article that GGA and PBE are some different approximations used simultaneously in the calculations. In reality, GGA is an approach considering electron density non-homogeneity, while PBE functional (not function) is one of the ways how this approach is presented mathematically.

Line 72: “single nanotube supercell”. The authors considered two-dimensional surfaces, but not nanotubes.

Line 91-92: “The bond lengths of Ge atom and Se atom in GeSe…”. It must be “The bond lengths between Ge and Se atoms in GeSe…”.

Line 95-96: “The CO gas molecule is a two-dimensional planar structure…”. A two-atomic molecule is not planar, it's linear.

And there are many such errors in the article.

2) The second problem is related to the research method and the interpretation of the corresponding results. The authors use the concept of molecular orbitals to explain the change in the band gap in the materials under consideration during the adsorption of CO and CH4 and, as a consequence, the change in conductivity.

First, it is not clear why the authors use the concepts of orbitals when the calculations were periodic and it is more correct to speak of a band structure.

Second, the authors present the band gap values (the difference between the HOMO and LUMO energies) for compounds of the considered materials with adsorbed gas molecules. To conclude the change in electrical conductivity, it is necessary to compare the band gap values of the material before and after the adsorption of analyte molecules. However, there are no data on the width of the band gap before the adsorption of methane and carbon monoxide in the article.

Third, and most importantly, the idea of using silver atoms on the GeSe surface is probably meaningless, and the authors show this, although they write about the opposite. The band gap of a GeSe monolayer is slightly more than 1 eV (https://doi.org/10.1063/1.4931459). The authors indicate that in the case of silver atoms and gas molecules on the surface of this monolayer, the band gap is in the range of 0.025-0.053 eV (Table 3, Figure 8). Obviously, this is actually not the band gap of the entire system, but the energy difference between the impurity levels inside the band gap. The boundaries of the band gap are formed by the orbitals of the Ge and Se atoms. The impurity levels are formed by the orbitals of silver atoms and gas molecules. Taking into account that the authors consider a supercell containing 32 Ge atoms and 32 Se atoms, the concentration of silver atoms is too low for any changes in the energies of the orbitals of silver atoms during the binding of gas molecules to affect the electrical conductivity of the entire structure.

Round 2

Reviewer 2 Report

OK now. Accept

Reviewer 3 Report

Authors have revised the document according the comments in the previous review. Threfore, now it can be accepted.

Reviewer 4 Report

Despite the fact that the authors made some corrections to the manuscript, the key problems in the choice of objects and methods of research and interpretation of the results have not been resolved.

As far as I understand, the vertical dotted line in Figures 3, 5, and 7 indicates the Fermi level. From this point of view, the band gap of all considered structures with or without gas molecules is equal to or close to zero, since the value of TDOS at the Fermi level is not equal to zero. Then it is not clear what band gaps, the values of which are given in Table 3, are in question.

Suppose that the authors have not normalized the energy values in Figures 3, 5, and 7 to the Fermi energy (Ef). However, this is usually done in practice and the values of E-Ef are given on the x-axis so that Ef corresponds to zero. Consider Figure 7a1. I do not see a significant difference in the plots of the densities of states of the Ag-GeSe and Ag-GeSe/CH4 systems in the range from -6 to 1 eV, where, in theory, there should be a band gap. At the same time, the authors indicate in Table 3 that the band gap of the Ag-GeSe system is 1.224 eV, and Ag-GeSe/CH4 is 0.049 eV! That is, the authors point to a significant change in the band gap during the adsorption of a methane molecule by the Ag-GeSe system, but, considering Figure 7a1, there is no significant change there. A similar statement applies to all other structures considered.

The authors, interpreting the changes in the density of states during the adsorption of a CO molecule in lines 144-151, indicate that these changes occur in the energy range from -12 to -10 eV, which is very far from the Fermi level. That is, the authors themselves do not note any changes at the Fermi level. A similar situation is observed in lines 179-189, where they are talking about changes in the densities of states in the energy range from -7 to -5 eV in the case of adsorption of the CH4 molecule. And since there are no changes within the band gap (if it is there after all), then there will be no significant change in the band gap itself.

In section "3.4. Molecular orbital theory analysis of gases adsorption on Agn-GeSe", the authors, using the concept of molecular orbitals, try to explain the observed changes in the band gap in the objects under study during the adsorption of gas molecules. If we assume that such changes exist, although I indicated the opposite above, the expediency of using the GeSe surface is questionable. Judging by the results of the authors, during the interaction of CO and CH4 molecules with silver clusters, there is a change in HOMO and LUMO, which are mostly localized on silver atoms, and this change does not affect the entire electronic structure of the Agn-GeSe system. Then it turns out that with the same success one can consider any other surface instead of GeSe to obtain a similar result.

In addition to what has been said, one should also point out the presence of a large number of uncorrected errors in the text. For example, one can observe the following contradiction. In lines 182-183 the authors indicate "Therefore, the conductivity increases after CH4 adsorption", however in lines 218-220 we already see "After Ag-GeSe adsorbs CH4, the band gap increases significantly, which makes the conductivity of the system decrease significantly ...". At the same time, it should be noted that according to the authors’ data, Eg in the case of, for example, Ag-GeSe is 1.224 eV, and in the case of Ag-GeSe/CH4 it is 0.049 eV, so Eg decreases, although here the authors, interpreting the results, indicate that it increases.

Based on the above, I believe that there are errors in the work, both in the choice of the object of study and in the use of methods and interpretation of the results. In this regard, I recommend rejecting the article from consideration regarding the possibility of publication.